# Extraction and Processing of Bioactive Phytoconstituents from Widely Used South African Medicinal Plants for the Preparation of Effective Traditional Herbal Medicine Products: A Narrative Review

**DOI:** 10.3390/plants14020206

**Published:** 2025-01-13

**Authors:** Sphamandla Hlatshwayo, Nokukhanya Thembane, Suresh Babu Naidu Krishna, Nceba Gqaleni, Mlungisi Ngcobo

**Affiliations:** 1Traditional Medicine Laboratory, University of KwaZulu Natal, Durban 4041, South Africa; thembane@mut.ac.za (N.T.); gqalenin@ukzn.ac.za (N.G.); ngcobom3@ukzn.ac.za (M.N.); 2Department of Biomedical Sciences, Mangosuthu University of Technology, Durban 4026, South Africa; 3Institute for Water and Wastewater Technology, Durban University of Technology, Durban 4000, South Africa; sureshk@dut.ac.za; 4Africa Health Research Institute, Durban 4013, South Africa

**Keywords:** medicinal plants, extraction, processing, bioactive phytoconstituents, potency

## Abstract

Medicinal plants are sources of crude traditional herbal medicines that are utilized to reduce the risk of, treat, or manage diseases in most indigenous communities. This is due to their potent antioxidant and anti-inflammatory effects. It is estimated that about 80% of the population in developing countries rely on herbal traditional medicines for healthcare. This signifies the need for traditional herbal medicines, which are polyherbal formulations prepared by traditional health practitioners. This review examines preparatory steps to extract bioactive phytoconstituents and post-extraction processes to increase the potency of the extracted bioactive phytoconstituents. Achieving this will allow for the reduced use of plant materials and promote the sustainable use of the limited resource of medicinal plants, especially in our South African context. Electronic ethnobotanical books and online databases were used to find studies that focus on phytoconstituent extraction and post-extraction processing to enhance the potency of the extracted bioactive phytoconstituents. Modification of the extracted bioactive phytoconstituents to synthesize daughter compounds facilitates an enhancement in their potency and bioavailability. Based on the data collected through this review, the importance of understanding the properties of the targeted phytoconstituents is essential in selecting the required extraction method. This determines the quality and yield of extracted bioactive phytoconstituents.

## 1. Introduction

According to epidemiological studies, over the years, there has been an exponential increase in the prevalence of diseases related to inflammation and oxidative stress, predominantly in developing countries [1]. In most developing countries, medicinal plants are sources of crude herbal medicines, which are utilized to reduce the risk of, treat, or manage acute and chronic diseases related to inflammation and oxidative stress [2].

This is due to the potent antioxidant and anti-inflammatory effects that medicinal plants possess due to the presence of various bioactive phytoconstituents [2]. In developed countries, treatment of diseases is achieved via preventative measures and synthetic drug administration coupled with monitoring records of the patient’s health [3].

In underdeveloped and developing countries, patients sometimes rely on traditional herbal medicines from traditional health practitioners (THPs) for primary healthcare [4].

It is estimated by the World Health Organization (WHO) that about 80% of the population in developing countries relies on traditional herbal medicines [5]. Such statistics validate the belief that traditional herbal medicines are utilized by a significant fraction of the world’s population [6]. In developed nations, the usage of traditional herbal medicines is also a fast-growing phenomenon as some synthetic drugs have been reported to result in deleterious side effects [6,7]. This has resulted in traditional herbal medicines increasingly becoming more accepted and popular in developed societies [6]. The formulation of traditional herbal medicines utilizes a number of medicinal plants to produce a polyherbal mixture [8]. According to the WHO, traditional medicine is defined as “health practices, approaches, knowledge and beliefs incorporating plant, animal and mineral based medicines, spiritual therapies, manual techniques, and exercises, applied singularly or in combination, to treat, diagnose and prevent illnesses or maintain well-being” [9].

More than 35,000 medicinal plant species have been identified and reported to be utilized for medical purposes, but only 121 bioactive compounds have been studied [10]. This signifies the importance of determining suitable pre-extraction processing and extraction methods to recover more bioactive phytoconstituents from medicinal plants. Bioactive phytoconstituents may be degraded due to the use of extraction methods, such as decoction and infusion, that are widely used by THPs as they lack selectivity [11,12,13]. Several natural bioactive compounds have been sourced from various medicinal plant species based on their traditional applications for health benefits. This has resulted in more than 25% of clinical conventional medications originating from medicinal plants [14]. This signifies that medicinal plants are pivotal in meeting the healthcare needs of people [6,7,14]. Scientific evaluation of various traditional herbal medicines and their constituent medicinal plants supports their traditional uses, which has led to the development of novel drugs [14,15]. Furthermore, the presence of bioactive phytoconstituents in traditional herbal medicines confirms the indigenous knowledge held by THPs and their indigenous communities [14,16]. Bioactive phytochemical constituents found in medicinal plants, such as flavonoids, phenolics, terpenes, coumarins, and saponins, are reported to exhibit numerous therapeutic effects. These assist in reducing the risk of multiple diseases, including inflammation, oxidative stress, and diabetes mellitus (DM) [17,18,19].

Herbal medicine products can be used as dietary supplements, nutraceuticals, or traditional medicines [20,21]. There are over 3400 plant species in southern Africa that are used for medicinal purposes [22]. Out of the 3400, 2062 medicinal plants have been reported as used in traditional herbal medicine preparations and/or traded in herbal markets that supply THPs and local communities [23,24,25]. This is in line with reports that have indicated that in South Africa, approximately 70,000 tonnes of plant material is harvested annually [26,27,28]. Plant material harvested from wild resources is considered more potent because of the presence of a wider range of bioactive phytoconstituents. The presence of bioactive phytoconstituents manifests as a protective effect against various stimuli in wild environments. Therefore, the presence of bioactive phytoconstituents may not be expressed as much in medicinal plants cultured in vitro [29]. Amid the growing number of diseases and patients that require traditional herbal medicines, the demand for medicinal plant material is also growing. However, a wide range of medicinal plant species are showing signs of unsustainable harvesting, seasonal irregular supply, and scarcity in some habitats [30]. Apart from use as traditional herbal medicines, threats to the sustainability of the medicinal plant resource base are further exacerbated by habitat destruction due to urbanization and industrialization [31]. This further threatens the sustainability of medicinal plants, which are an integral part of traditional herbal medicines [30]. In the African context of preparing traditional herbal medicines, combining plants in herbal mixtures is widely practiced by THPs. Polyherbal remedies may be traditionally prepared as infusions, decoctions, and macerations, depending on the target bioactive phytoconstituents being extracted [12,13]. Because a certain medicinal plant may be a constituent of a number of traditional herbal medicines, its wild harvesting may likely reach exploitive and unsustainable levels. Taken together with habitat destruction, which leads to deforestation, this provides evidence that the limited natural resource of medicinal plants, including the ones that are widely used in the preparation of traditional herbal medicines in South Africa, is declining. These include the decreased availability of *Adansonia digitata* L., *Agrimonia eupatoria* Krylov, *Aloe ferox* (Mill.), *Aspalathus linearis* (Burm.f.) Dahlg., *Eucomis autumnalis* (Mill.) Chitt., *Harpagophytum procumbens* (DC. Ex Meisn.), *Pelargonium sidoides* DC., *Plumbago auriculata*, *Psidium guajava* L., *Sclerocarya birrea* (A. Rich.) Hochst., and *Sutherlandia frutescens* (L.) R.Br., to name a few [13,32,33,34,35,36]. Based on this, it has been estimated that the global population loses at least one potential major drug that can be sourced from natural wild resources every 2 years [14]. Due to a high demand for herbal medicinal products that promote health in Africa, there is a need to study and identify the most effective pre-extraction processes and extraction methods to preserve bioactive phytoconstituents from medicinal plants. Additionally, the introduction of post-extraction processes to improve the efficacy of extracted bioactive phytoconstituents is important. The conservation and sustainable use of medicinal plants has been studied extensively in the field of biological sciences. Various sets of recommendations have been compiled regarding the conservation of medicinal plants. Therefore, this review aims to explore the extraction and processing of bioactive phytoconstituents from widely used South African medicinal plants for the preparation of effective traditional herbal medicine products. Achieving this will ensure that fewer plant materials are utilized when preparing traditional herbal medicines to reach the threshold potency of these medicinal products.

## 2. Literature Search Strategy

The literature search encompassed electronic databases such as PubMed, Scopus, Science Direct, Web of Science, and Google Scholar, covering studies from the year 2000 up to June 2024. Keywords such as medicinal plants, traditional herbal medicine, phytoconstituents, extraction methods, derivative synthesis, and nanotechnology were used to search for relevant articles. Studies were screened independently for relevance to bioactive phytoconstituent extraction and post-extraction processing. After full-text retrieval and exclusion of duplicates, insufficient data, and non-English publications, the search identified 301 studies for possible inclusion. Studies that focused on bioactive phytoconstituents extracted from non-plant material were excluded, and 286 studies met the inclusion criteria included in this review.

### 2.1. Pre-Extraction Processing of Medicinal Plants

Prior to the extraction of bioactive phytoconstituents from the harvested medicinal plants or parts of the plant (roots, leaves, flowers, stem, or bark), they are normally sun- or oven-dried. Drying facilitates preservation, as it limits and restricts bacterial activity and fungal growth, which are usually found in moist and watery environments [37]. Drying also arrests the potential activity of active enzymes that may be present in the fresh plant material. This prevents enzymatic degradation that may occur on the plant material and its bioactive phytoconstituents [38]. Drying is followed by cutting into smaller pieces or grounding the plant material to increase the surface area for the extractant. This improves the contact of the powdered plant material with the solvent during the bioactive phytoconstituent extraction process [39]. The particle size of the plant material determines the degree of penetration of the solvent to extract bioactive phytoconstituents. Therefore, the yield of extracted bioactive phytoconstituents is directly dependent on the two abovementioned steps [40]. In addition to the above processes, in African traditional medicine (ATM) practice, these medicinal plants are combined into polyherbal formulations. Extracting bioactive phytoconstituents from polyherbal formulations presents a wider array of phytoconstituents due to the number of medicinal plants utilized. This phenomenon presents a new class of bioactive phytoconstituent extraction compared to singular plant extractions that have been reported in the majority of laboratory settings. In research related to the utilization of bioactive phytoconstituents, the extraction process is pivotal, as it is where the preservation of the bioactive phytoconstituents being isolated takes place [41]. Since medicinal plants usually consist of several bioactive phytoconstituents, this presents a need for the development of extraction techniques/methods that can preserve bioactive phytoconstituents [41].

The parameters for selecting an appropriate extraction method include the following:i.Botanical verification of the medicinal plant or its parts where bioactive phytoconstituents are to be extracted [42].ii.The age of the plant, season, and exact location of harvest, including habitat [43].iii.The nature of constituents:
(a)Whether or not the bioactive phytoconstituents require polar or non-polar solvents.(b)If the bioactive phytoconstituents are heat sensitive or not (as some extraction methods are performed at high temperatures).(c)The duration required for the optimal extraction of bioactive phytoconstituents is vital as a shorter than required extraction time would result in an incomplete extraction. Or, if the duration of the extraction time is exceeded, unwanted phytoconstituents may also be extracted.(d)The concentration and drying procedures should ensure the safety and stability of the bioactive phytoconstituents.(e)Analytical parameters of the final extract, such as thin layer chromatography (TLC) and high pressure liquid chromatography (HPLC), should be documented to monitor the quality of different batches of the extracts [44,45].

### 2.2. Extraction Methods

#### 2.2.1. Maceration

The term “maceration” denotes softening. The process of maceration extraction is a solid (plant material)–liquid (solvent) extraction process. The grounded plant materials are soaked in the solvent in a container that can be closed, and regular shaking and agitation is applied [46]. The plant material–solvent mixture is allowed to stand at room temperature for 2–7 days, depending on the bioactive constituents being extracted [47]. This process softens and breaks down the plant material cell walls, resulting in the release of cellular encapsulated bioactive phytoconstituents into the extractant solvent [48]. When the extraction is complete, the solvent is strained out, and the insoluble plant material is pressed to extract all the solvents embedded within the plant material. In some instances, pressing is replaced by filtration or centrifugation [49].

#### 2.2.2. Tisane/Infusion Extraction

Infusion extraction refers to a process used to extract plant material that is readily soluble. This material usually dissolves easily to release bioactive phytoconstituents when in contact with organic solvents [50]. During this extraction process, plant materials are soaked in a specific volume of either hot or cold solvent for approximately 15 min. This is followed by cooling, which takes approximately 45 min, and then followed by filtration [51]. This extraction method is a safe and effective process for crude drug extraction and is also recognized by the Indian Pharmacopoeia for the extraction of crude drugs [52].

#### 2.2.3. Decoction

A decoction is a water-based preparation used to extract active components from medicinal plants. The plant materials are boiled in a certain volume of water for a set period (15 min to 2 h), followed by cooling and filtration [51]. A decoction is best suited for extracting water-soluble, heat-stable, and hard bioactive phytoconstituents from plant materials [51]. Delicate plant parts such as leaves, roots, flowers, and tender stems are boiled for 15 min. Hard plant parts such as branches and tree bark can be subjected to boiling for up to an hour [51]. Herbal medicines produced via the decoction extraction method are mostly consumed orally [53]. The decoction method of extraction of phytoconstituents from medicinal plants is still largely used by THPs when formulating traditional herbal medicines (Figure 1) [54]. This method of extraction lacks selectivity (especially water-insoluble compounds), produces lower yields, and consumes large volumes of water. Therefore, it presents a safety concern and environmental risk [53].

#### 2.2.4. Soxhlet Extraction

Soxhlet extraction involves the use of a Soxhlet apparatus for the extraction of bioactive phytoconstituents from herbs. In this extraction method, the plant material is repeatedly subjected to a warm solvent to provide a higher extraction yield [56]. The plant material is placed in a thimble holder (plant material holder) that is repeatedly supplied with a solvent from the distillation flask. The solvent is placed in the distillation flask, where it is subjected to heat. Heating results in the solvent being delivered as a vapor into the distillation arm to the plant material in the thimble [57]. Extraction is initiated via contact of the plant material and the condensed solvent that was vaporized. Soxhlet extraction takes place in a continuous cycle as the solvent is recirculated through the sample. As the condensed solvent almost reaches full capacity on the thimble, a siphon aspirates the solvent from the thimble back into the flask (Figure 2) [58]. The bioactive phytoconstituents being extracted from the plant material are isolated via rotary evaporation [59]. Soxhlet extractions are limited by the time required for the extraction process and the large volumes of solvent required for optimal extraction. Such extractions are performed at the boiling point temperature of the solvents. Hence, if the boiling point of the bioactive phytoconstituents being extracted is lower than that of the solvent, the bioactive phytoconstituents being extracted may be susceptible to thermal decomposition. This may result in less biological activity or a highly reduced potency of the herbal medicine, as the bioactive phytoconstituents would not be preserved [60]. Soxhlet extraction takes about 6–48 h and is performed at about 65–100 degrees Celsius [61].

New extraction methods with improved efficiency and selectivity are replacing traditional methods of extraction in laboratories and industrial settings. This is due to shortfalls such as the lack of high-performance and reliable extraction techniques and methodologies for establishing the purity and standard for herbal medicines [63]. Such factors mean that herbal medicines require robust extraction methods in order to produce solutions for a global healthcare market. The traditional method of solvent extraction of bioactive phytoconstituents from plants is based on the suitability of solvents and the use of heat and/or agitation, which improves the solubility and transfer of the target bioactive phytoconstituents [40]. Usually, traditional techniques require a longer extraction time, which poses a risk to the thermal susceptibility of the bioactive phytoconstituents being extracted [64]. Novel extraction methods, including microwave-assisted extraction (MAE), supercritical fluid extraction (SCFE), accelerated solvent extraction (ASE), subcritical water extraction (SWE), and ultrasound-assisted extraction (USE), have drawn significant research attention in the past two decades [39]. If these techniques are explored scientifically, they can provide an efficient extraction technology for ensuring the quality of herbal medicines worldwide [64]. The limiting factor in our settings is that there are no studies that have been performed using these new extraction methods in African traditional formulations to assess their suitability to extract bioactive phytoconstituents for such formulations.

#### 2.2.5. Accelerated Solvent Extraction

Accelerated solvent extraction, sometimes called pressurized solvent extraction or pressurized liquid extraction, is an extraction method that offers advantages such as reduced extraction times and solvent consumption and increased extraction yields in comparison to Soxhlet and decoction extraction methods [65]. ASE equipment consists of pressure and heat settings, which facilitate extraction via reduced solvent viscosity (Figure 3). ASE is performed at temperatures that are higher than the boiling point of the extractant solvent. Pressure helps maintain the solvent in liquid form at elevated temperatures during the extraction process [66]. The temperature and pressure in this system are set to be constant for the duration of the extraction process. The plant material and the solvent are placed in a closed container inside the pressure vessel [67]. The container is connected to a thermocouple that detects temperature fluctuations in the sample container. Therefore, if the temperature changes, heating or cooling will take place to maintain the set temperature of the sample container [66]. The pressure, on the other hand, is controlled by the pressure relief valve. Whenever there are pressure fluctuations, the pressure relief valve will be opened to prevent pressure from building up or the pressurization system will apply pressure to the vessel to increase the pressure [67]. The ASE technique consists of mixing plant material with a solvent, followed by the extraction of the target bioactive phytoconstituents, and, lastly, the removal of solid (insoluble) plant material from the supernatant via filtration [68]. ASE is performed at high temperatures (50 to 200 °C) and pressures ranging from 10 MPa to 15 MPa. This pressure range is the highest when compared to other extraction methods. The ultra-high pressure and temperatures applied in the ASE method increase the solubility of plant materials, resulting in shorter extraction times [69]. Hence, ASE is performed at temperatures greater than the boiling point of the solvent to improve the extraction kinetics via the disruption of solvent–bioactive phytoconstituent interactions (i.e., van der Waals forces, hydrogen bonding, dipole interactions), increased molecular motion of solvent molecules, and enhanced phytoconstituent solubility in the extraction solvent as a result of the elevated temperature [70]. An increased temperature also accelerates the extraction kinetics by keeping the viscosity of the solvent low, which allows for extensive penetration of the plant material. This results in the rapid diffusion of phytoconstituents into the solvent medium, which consumes less time and solvent [65,70]. The extraction solvents used are selected based on their compatibility with the phytoconstituents being extracted and post-extraction methods (purifying the phytoconstituent from the extractant solvent). Bioactive phytoconstituents found in herbal medicines are mostly susceptible to thermal degradation. Therefore, the temperatures required for ASE may result in thermal degradation and, eventually, the loss of the biological activity of the phytoconstituents being extracted [60]. Extract impurities due to the high pressure applied during extraction also present a setback for the ASE method. This tightly embeds the extraction solvent into the extract, therefore slightly affecting the purity of the extract [71]. The extraction time in PLE varies from 5 to 20 min and the extraction efficiency varies between 95 and 100% when compared to traditional extraction methods [72].

#### 2.2.6. Supercritical Fluid Extraction

Broadly, the SCFE process can be divided into two: the extraction process of bioactive phytoconstituents from the plant material followed by the removal of the solute from the solvent. During this process, the solvent is subjected to heat and pressure to induce its critical state before extraction commences. The SCFE system consists mainly of a chiller used to cool a solvent, a solvent pump, an extraction column, separators, temperature regulators, a pressure regulator, a back pressure regulator, and an oven (Figure 4) [74]. The solvent pumps circulate the solvent throughout the system. The extraction column is where the sample (plant material) to be subjected to extraction is placed. The separators collect the final product (extracted bioactive phytoconstituents), and the temperature regulators are used to adjust the temperature during the extraction process. The pressure regulator maintains pressure in the system within the required ranges of a particular extraction. The oven is used to keep the extraction column above the critical temperature of the extraction fluid [74,75]. A fluid is in its critical state when its temperature and pressure are above its respective critical values [75,76]. During the process of SCFE, the solvent properties are an intermediate phase between a gas and a liquid. This induces a solvent power that possesses both liquid- and gas-like viscosity, resulting in enhanced bioactive phytoconstituent transfer from the plant material [77]. As a function of pressure and temperature, changes in the density of the fluid in its supercritical state permit excellent solvating power, allowing for selective extractions by adjusting both the pressure and temperature according to the optimal settings of the bioactive phytoconstituents being extracted [75,77]. The extraction mechanism of SCFE is categorized into (i) diffusion of the supercritical solvent into the plant material [78] (ii) followed by the bioactive phytoconstituents being deeply embedded into the extractant solvent as a result of the plant material penetration; (iii) the third stage is the removal of the extracted bioactive phytoconstituents from the solvent–bioactive phytoconstituent fluid mixture, followed, (iv) lastly, by the recovery of the bioactive phytoconstituents from the solvent–bioactive phytoconstituent fluid mixture via decompression [79]. The specificity of SCF extraction relies on the physicochemical properties of the bioactive phytoconstituents being extracted, which can be modulated by an increase in pressure and temperature. Viscosity and diffusivity are factors that affect the penetration of the solvent into the plant materials. The decreased viscosity and increased diffusivity of the solvent facilitate the penetration of the solvent into solid materials [78]. This results in increased bioactive phytoconstituent transfer from the plant material and reduced extraction times in SCFE [78,79]. This is followed by the removal of extracted bioactive phytoconstituents from the solvent, which takes place in the separator. Once this is performed, the solvent is recirculated for re-use, where it is induced to its supercritical state to continue with the process of extraction [80]. The extracted compound can be collected at the bottom of the separator [81]. The SCFE process takes roughly 30–60 min to complete, depending on the bioactive phytoconstituents being extracted [50].

#### 2.2.7. Microwave-Assisted Extraction

Microwaves are non-ionizing waves/radiations that consist of electric and magnetic fields. The electro and magnetic fields of microwaves oscillate perpendicularly to each other within a frequency range of 300 MHz to 300 GHz, located between the X-rays and infra-red rays in the electromagnetic spectrum [83]. Microwave-assisted extraction (MAE) is an extraction process that uses a solvent such as water or alcohol to extract the bioactive phytoconstituents from medicinal plants [84]. During MAE, the microwave transfers energy to the solvent and plant material. This facilitates extraction due to changes in the plant material cell structure caused by electromagnetic waves [85]. The microwave energy is delivered directly to the medicinal plant cellular particles through interactions with the electromagnetic field. The energy is absorbed by the plant material that has been penetrated by the solvent. The microwave energy is converted to heat energy [84,85]. Heating may cause liquid vaporization within the cells, which may rupture the cell walls to facilitate the penetration of plant material by the extractant, resulting in greater extraction yield (Figure 5) [86]. Devices utilized for MAE are closed and focused extraction vessels. In a closed extraction vessel, MAE takes place under regulated pressure and temperature, while in a focused extraction vessel, MAE is performed at atmospheric pressure. Therefore, a closed extraction vessel is recommended for bioactive phytoconstituents that require elevated temperature and pressure [87]. The principle of heating using microwaves is based on its interaction with a solvent, facilitated by ionic conduction and dipole rotation, which happen simultaneously [88]. Ionic conduction refers to the migration of negatively and positively charged molecular bodies as a result of the driving force exerted by the electric field [89]. The resistance caused by the solvent to the migrating ions and electrons generates friction, which eventually warms up the solvent. This phenomenon of ionic conduction occurs when the frequency is ~2450 MHz [90]. Dipole rotation refers to the realignment of the negatively and positively charged poles of the molecules being realigned due to the driving force exerted by the electric field. This results in collisions between dipoles and surrounding molecules, thus generating heat. Polar solvents have greater heat-producing abilities when subjected to microwaves, compared to lesser polar solvents [91]. The heat production efficiency of different solvents, when subjected to microwaves, depends on the dissipation factor, which is the measure of the ability of the solvent to absorb microwave energy and pass it on as heat to the surrounding medicinal plant material where bioactive phytoconstituents are extracted [90,91]. The end result of this entire process is the breakdown of the plant material cell wall, leading to the release of bioactive phytoconstituents [85]. The duration of MAE can last up to 20 min when performed at a temperature range of 120–140 degrees Celsius, depending on the bioactive phytoconstituents being extracted [92].

#### 2.2.8. Sonication/Ultrasound-Assisted Extraction

The sonication/ultrasound extraction method can be carried out via the use of an ultrasound probe or an ultrasound bath, whereby a transducer is a source of ultrasound waves [94]. Apart from sonication, a pressure of 50 MPa and a 40–60 degree Celsius heat application are required to perform this extraction [95]. The sonication/ultrasound-assisted extraction technique employs ultrasonic waves found between the frequency range of 20 kHz–10 MHz, which are audio and microwaves (Figure 6). The ultrasound wave range is presented by power ultrasound (20 kHz–100 kHz) and diagnostic ultrasound (100 kHz–10 MHz) [96]. The ultrasound probe system is preferable in the extraction industry. This is due to the power of intensity that it can deliver through a small surface area, which is the tip of the ultrasound probe [97]. Ultrasound extraction using a probe requires the probe to be immersed into the solvent–plant material mixture and operated at a frequency of 20 kHz. Power ultrasound is used for bioactive phytoconstituent extraction and food processing applications [98]. Diagnostic ultrasound is used in clinical settings for diagnostic instruments [99]. The mechanical effect from the ultrasound increases the surface area of contact between solvents and plant material. This allows the permeability of the plant material cell wall by the solvent via bubble formation, followed by bubble growth and, eventually, the rupture of the plant material cellular structures after the bubble has collapsed [100]. The extraction of bioactive phytoconstituents is achieved based on acoustic cavitation. Acoustic cavitation is a process that consists of the formation, growth, and collapse of bubbles when ultrasound waves travel through the extractant solvent following irradiation with an ultrasound wave at a frequency greater than 20 kHz [101]. Once bubbles are formed, their growth depends on frequency, pressure, and bubble radio [102]. Bubble growth follows bubble formation, which can be explained by coalescence or the rectified diffusion phenomenon. Coalescence refers to two bubbles combining to form one bigger bubble. Rectified diffusion is defined by single bubble growth as a result of pressure gradient differences between the outer and the inner regions of bubbles [102]. The end result is bubbles collapsing at the end of acoustic cavitation and producing the cavitation effect. This leads to the formation of microcracks on the surface of solids due to the release of high energy when the bubble is collapsed [100]. This effect results in improved solvent penetration into the plant material and the release of cellular contents to the solvent. This facilitates bioactive phytoconstituent transfer from the plant’s cellular structures [103].

From the comparison of methods for extracting various phytochemicals, it can be concluded that researchers have been conducting research to discover methods that could result in higher extraction yield, possess better selectivity, use less solvent and energy, have shorter extraction times, and are environmentally friendly processes. Table 1 below is a comprehensive summary of these extraction methods, including their advantages and disadvantages, and the suggested inputs to improve these methods.

## 3. Post-Extraction Methods for Enhancing the Potency of Phytoconstituents

### 3.1. Derivatives Synthesis

Based on the structural and pharmacokinetic characteristics of the bioactive phytoconstituent (parent compound), derivatives are synthesized to improve the efficacy of the parent compound [200]. Properties such as low solubility and poor bioavailability associated with herbal medicines limit the application of traditional medicine products in clinical settings [201]. Therefore, strategies toward the improvement of their properties are in urgent demand.

Based on the structural and pharmacokinetic characteristics of the extracted bioactive phytoconstituent (parent compound), derivatives are synthesized to improve the desired efficacies [200]. Synthesized derivatives exhibit their potency via improved physicochemical, biopharmaceutical, or pharmacokinetic properties of pharmacologically active compounds, thereby optimizing absorption, distribution, metabolism, excretion, and toxicity properties for potential drug candidates, which finally improves therapeutic indices of the parent drugs [200,201]. A prodrug is a chemically modified version of a pharmacologically active agent that elicits its desired pharmacological effects due to chemical modification(s) to the parent compound [202]. A prodrug undergoes chemical and/or enzymatic biotransformation in a regulated or predictable manner prior to exerting its biological effects [203]. Specifically, the conjugation of parent drugs with different functional groups (phosphoric acids, sulfuric acids, amino acids, polymers, or sugars) is employed to improve aqueous solubility, bioavailability, and pharmacokinetics [204]. The conjugation of non-ionized functional groups (e.g., alkyl or aryl esters) has been applied to enhance lipophilicity and oral or topical absorption of parent drugs [205]. It is estimated that currently, about 10% of world-wide marketed drugs can be classified as prodrugs [206]. These include 2-fluoroadenosine (F-ara-A), which has clinical use as an anti-neoplastic agent. However, it is difficult to formulate because of its lipophilicity. Therefore, fludarabine phosphate (2F-ara-AMP), which is a prodrug that is rapidly dephosphorylated to give fludarabine (F-ara-A), was synthesized [207]. Another example is the anti-Parkinson’s agent L-DOPA. This prodrug increases the efficacy of dopamine, which does not efficiently cross the blood–brain barrier. However, the prodrug of dopamine, L-DOPA, enables the uptake of dopamine into the brain [208].

### 3.2. Nanoparticle Synthesis

The unique features of nanoparticles offer various therapeutic advances in the field of drug delivery due to their potential to improve the clinical efficacy of bioactive phytoconstituents obtained from medicinal plants [209]. Nanoparticulate formulations in herbal medicines improve drug delivery by introducing alternative routes of drug administration, which have been reported in hydrophilic and hydrophobic drugs, proteins, vaccines, and biological macromolecules [210]. Some of the challenges of most drug delivery systems related to herbal medicines include poor bioavailability, in vivo stability, solubility, intestinal absorption, sustained and targeted delivery to specific receptors, therapeutic potency, side effects, and plasma fluctuations of drugs that either fall below the minimum effective doses or exceed the safe therapeutic concentrations [211,212,213]. However, nanotechnology drug delivery systems overcome these challenges through the development of drug delivery systems that can be applied to reformulate existing drugs. This has the advantages of extending the products’ shelf life, and potency, as well as increasing the safety profile of the drug, which may significantly contribute to patients’ adherence to medication and, ultimately, a reduction in healthcare costs [214]. For instance, nanotechnology has been successfully incorporated into therapeutics used for cancer treatment [7,8]. This has led to the development of drugs such as Dox-il, DaunoXome, Myocet, DepoCyt, Marqibo, and Onivyde [215]. These drugs have also been granted approval by the Food and Drug Administration (FDA) for treating diseases such as COVID-19 [216]. Nevertheless, to be clinically relevant, nanoparticle-based therapies must be produced through techniques that ensure stability during storage, compatibility with sterilization, quality control, and regulatory compliance [217]. An example of such challenges is that during preparation or storage of lipid-based nanoparticles, triglycerides may convert from their α-form to the β-form, leading to the formation of aggregates that result in drug leakage [218]. Industrial mass production also requires sterilizing. This presents a limit due to destabilization that may result from conventional sterilization methods [218].

### 3.3. South African Medicinal Plants with Phytochemicals That Can Be Enhanced Through Post-Extraction Processing

African traditional medicines in South Africa are prepared via the utilization of various medicinal plants as polyherbal formulations. According to traditional knowledge, the use of medical plant formulations facilitates reducing the toxic effects of some medicinal plants while also eliciting synergistic effects. A variety of plant material preparation methods are used to prepare traditional medicine for different types of ailments and diseases in the study area, as indicated in Table 2. Decoction, infusion, paste, and ash were recorded as the main methods utilized for the extraction of bioactive phytoconstituents [12,13].

## 4. Challenges for the Herbal Industry

Bioactive phytoconstituents are vital constituents of herbal products because bioactive phytoconstituents determine the safety and efficacy of herbal products [263]. Data on phytoconstituent processing can be difficult to obtain; particularly, the physical and chemical properties, including solubility values, partition coefficients, and heat transfer coefficients, are not available [264]. These critical data are required to develop an effective process model that can be integrated into process design methods [263,264]. The physiochemical property data can be obtained through either experimental studies or mathematical modeling of developmental drugs. Determining the physical and chemical properties of phytoconstituents can be challenging because herbal materials contain multiple phytochemical components, and each component contributes to these properties [263,265]. Reports on herbal extraction analysis using either GC or HPLC indicate that the bioactive phytoconstituents vary with the method and solvent applied during the extraction process [266]. In the case of THPs, who solely rely on wild resources for medicinal plant harvesting, this presents a challenge as there may be seasonal and habitat/environmental harvest variations from each batch. It is therefore expected that there will be batch-to-batch variations in the bioactive phytoconstituent contents between harvests, making it difficult to ensure consistency in their traditional herbal medicines. This has a huge impact on the use of wild-harvested medicinal plants to meet the healthcare needs of people, especially through the practice of traditional healing [267]. In addition to the aforementioned challenges, habitat destruction due to urbanization and industrialization poses threats to the practice of traditional healing due to deforestation, which eliminates a critical resource for the practice [268,269]. Thus, this further signifies the need to develop methods that can effectively optimize the processing, extraction, and post-extraction processing of target bioactive phytoconstituents and improve the potency of the extracted bioactive phytoconstituents. This would ensure that, with the limited supply of plant material, the maximal health benefits of medicinal plants would still be achieved.

### 4.1. Standardization of Traditional Herbal Medicines and Their Products in South Africa

The growing demand for traditional herbal medicines raises concerns about quality control, safety, efficacy, and reproducibility when formulating traditional herbal medicines [270]. The standardization of traditional herbal medicines and their products is crucial [271]. Standardization would involve monitoring and controlling various components of traditional herbal medicines. This includes product development, manufacturing, and distribution. Standardization will guarantee that traditional herbal medicines and their products meet standards and specifications set by a regulatory body [272,273]. However, the South African Health Products Regulatory Authority (SAHPRA) has established regulations for complementary and alternative medicines but has not yet formulated regulations for African traditional herbal medicines and their products. This hinders the local traditional herbal medicine industry from building practice standards, accurate packaging, and labeling of products for consumers to make informed choices about these products [274]. The steps required to achieve standardization include macroscopic analysis, which refers to the botanical identification and geographic origin of the constituent plant material. Microscopic analysis and chemical analysis involve the confirmation of moisture content, phytochemical composition, microbial contaminants present, heavy metal contaminants present, residual solvents present, and bioactive phytoconstituent identification via qualitative and quantitative analysis [275,276,277]. Standardization also includes establishing the best suitable extraction method(s), and process validation should be performed to ensure consistent and reproducible extraction results. These should be compared to established predetermined quality standards, including extraction efficiency, yield produced, and the presence of targeted bioactive phytoconstituents [278,279,280].

The standardization of traditional herbal medicines would require the establishment of a unified framework based on national guidelines and good manufacturing practices. The encouragement of multidisciplinary partnerships between THPs, scientists, and regulators would bring together both traditional knowledge and modern expertise [281]. It would also preserve the valuable wisdom of THPs and indigenous communities while exercising the highest standards of modern care [282].

### 4.2. Conservation of Medicinal Plants to Sustain the Production of Traditional Herbal Medicines

The International Union for Conservation of Nature (IUCN) Red List of Threatened Species records data concerning endangered plant species globally [283]. Following the identification of endangered plant species, different methods are adopted. In situ conservation is the most appropriate conservation approach for the preservation of species, including endemic species. In situ conservation preserves the original genetic and geographical centers of biodiversity, resulting in the conservation of ecosystems and biodiversity in their natural habitats [284]. Ex situ conservation, the conservation of biodiversity outside its natural habitats, is a widely adopted option for the preservation of rare and endemic species [285]. Ex situ conservation can also be achieved via conventional seed bank dry storage at −20 °C, mostly performed for plant germplasm [285]. However, some plant species cannot be preserved via these methods. Therefore, the introduction of biotechnology, in vitro propagation, slow growth preservation, and cryopreservation have contributed immensely to preserving endangered plant species conservation [286]. The introduction of these strategies is not to replace traditional conservation methods but rather to complement and improve the methods available [285,286].

## 5. Limitations

There was a substantial bias in the literature selection as this study was based on available and accessible data on the chosen search engines. Limitations associated with this study include its reliance on research lab-based setting reports on the various medicinal plant processing, extraction methods, and post-extraction processes discussed in this paper. This is biased against industrial-based data reports as there are far fewer reports on the industrial applications of different extraction methods. This indicates that there is a gap in the literature between the research laboratory applications and industrial applications of these methods. Another limitation is the missing reports on why THPs heavily rely on the decoction extraction method instead of other methods, as the conventional methods are widely researched. To overcome such limitations, it is essential that researchers collaborate with THPs to share knowledge of different extraction methods for medicinal plant preparation. This will not only enhance the work of THPs but also inform researchers about indigenous ways of extraction. This may further benefit researchers and THPs by producing their medicines in higher yields compared to the ones offered by current methods. Focusing on the exchange of knowledge between researchers and THPs may even bring about standardization of some extraction methods, which would advance herbal medicines production.

## 6. Conclusions

This review stresses the importance of refining extraction processes and exploring innovative post-extraction techniques to ensure the sustainable use of medicinal plants. By adopting a collaborative relationship between research and traditional practices, we can advance herbal medicine, making it more effective and accessible to meet the diverse healthcare needs of communities. Through these efforts, we can work towards a future where traditional healing practices are enriched by modern scientific advancements, safeguarding and enhancing medicinal plant resources for generations to come.

## 7. Future Perspectives

It is pivotal to research and develop extraction methods that are selectively compatible with target bioactive phytoconstituents being extracted, as higher yields may result in higher productivity of medications required to meet human needs for disease eradication. This necessitates the development of extraction techniques that require less solvent, energy, and time consumption. This would result in the elimination of the toxic and non-organic solvents used. Hence, protecting resources of the natural environment against environmental pollution is a global burden. Based on the type of work that is conducted in our laboratory, we aim to adopt some of these methods in our research settings to achieve the sustainability of traditional healing practice through innovative approaches to research and development. We will be working hand in hand with THPs to modernize the practice to provide traditional herbal medicines for the broader public.

## Figures and Tables

**Figure 1 plants-14-00206-f001:**
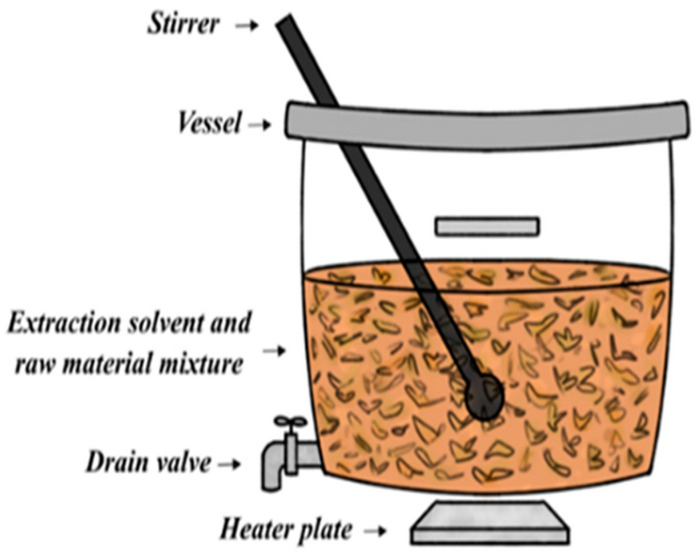
Diagram depicting the decoction extraction components [55].

**Figure 2 plants-14-00206-f002:**
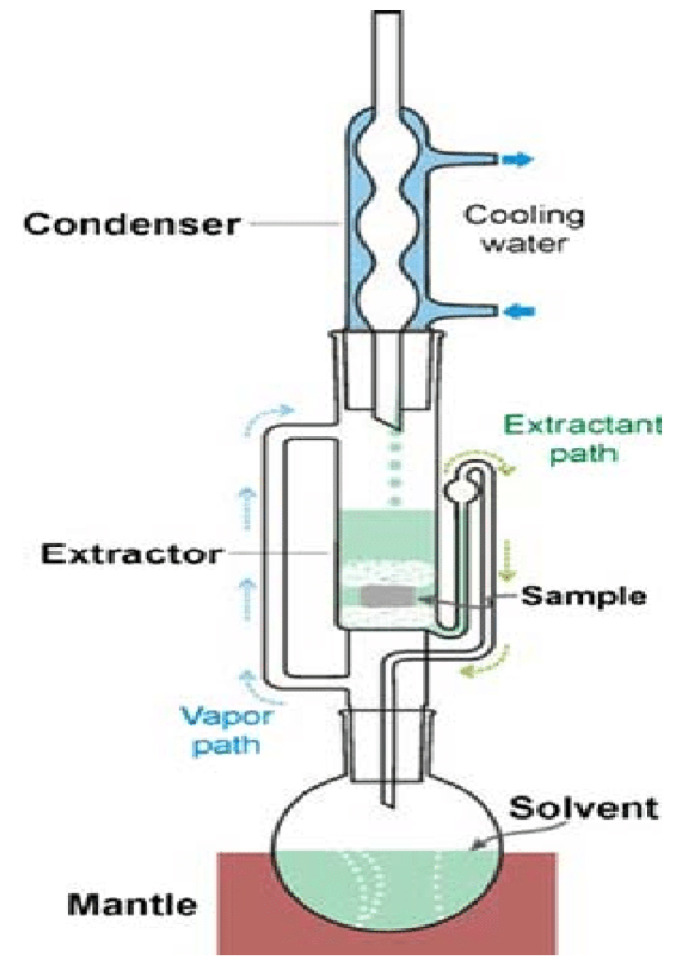
Diagram depicting the Soxhlet extraction apparatus [62].

**Figure 3 plants-14-00206-f003:**
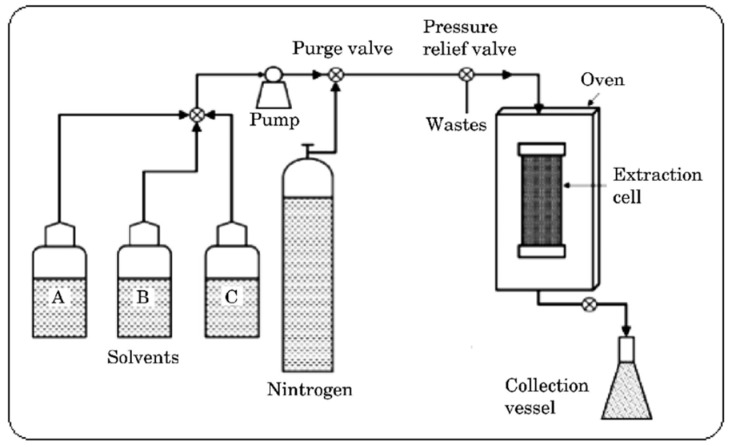
Diagram depicting the accelerated fluid extraction apparatus [73].

**Figure 4 plants-14-00206-f004:**
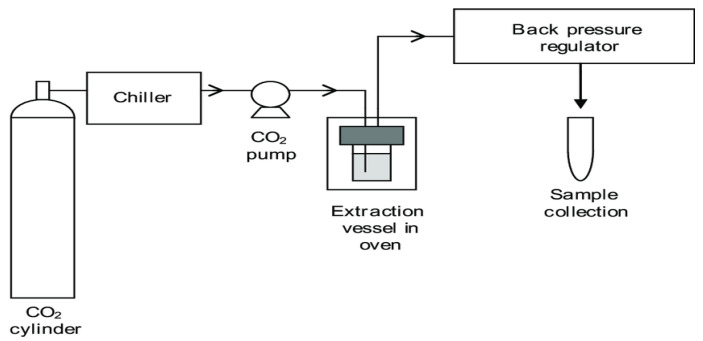
Diagram depicting supercritical fluid extraction [82].

**Figure 5 plants-14-00206-f005:**
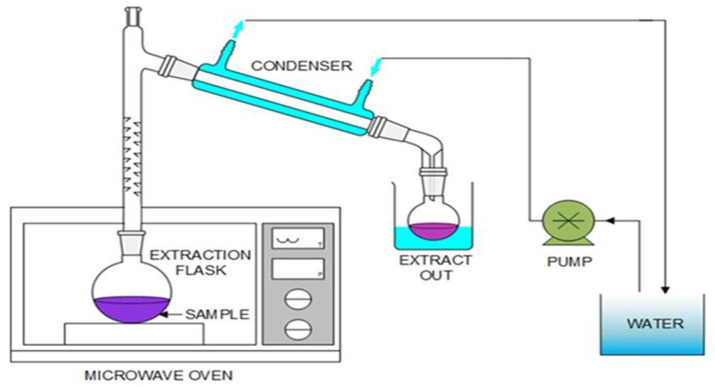
Schematic depicting a microwave-assisted extraction apparatus [93].

**Figure 6 plants-14-00206-f006:**
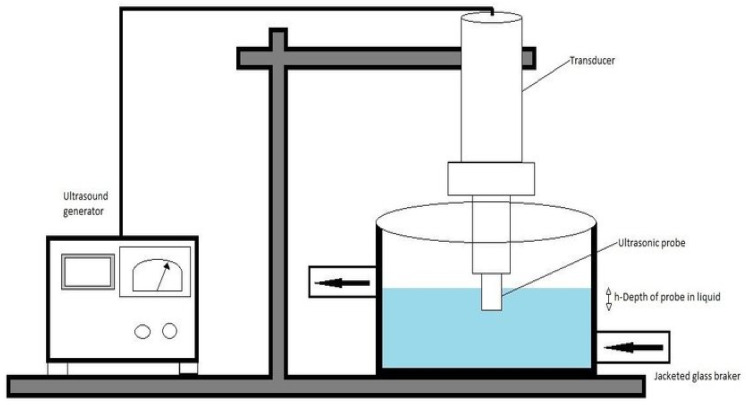
Diagram depicting a sonication-assisted extraction apparatus [104].

**Table 1 plants-14-00206-t001:** Medicinal plant extraction methods, including their advantages and disadvantages, and suggested inputs to improve these methods.

Extraction Method	Application	Advantages	Disadvantages	Suggested Inputs to Improve the Method
**Maceration**	This method is widely applied in wine making [10]. It is used to extract nonvolatile compounds in the pharma industry [48].	The method is simple, only requiring a vessel with a lid [105]. There is no need for a skilled technician to perform the extraction [106]. The process is heat-free, thus conserving energy [107]. This method is suitable for less non-hard plant materials [108].	Duration of the process is 2–7 days [47]. Consumption of large volumes of solvents compared to novel extraction methods [109]. Some solvents used are hazardous [92]. The process lacks selectivity [110]. Increased risk of microbial contamination, especially when water is used as the solvent [111]. Produces low extraction yield compared to novel extraction methods [110]. Suitable for nonvolatile substance extraction [48].	Due to the longevity the process, the addition of alcohol may be required to inhibit microbial development [112]. Introduction of a warm solvent to the extraction process [113].
**Infusion**	Infusion finds its application in tea preparation [114].	Best suited for delicate plant material such as petals, leaves, and flowers [115]. This extraction method is recognized by the Indian Pharmacopoeia for the extraction of crude drugs [52]. Does not require complex settings or expensive equipment for setup [116].	Effective in extracting readily soluble secondary metabolites [50]. Infusions do not accommodate hard plant materials such as bark and roots [50].	
**Decoction extraction**	Formulation of African traditional medicines, decoctions, concoctions, and teas [117].	Does not require complex settings or expensive equipment for setup [116]. Best suited for extracting bioactive phytoconstituents from hard plant materials such as bark and roots [50]. Thermo-stable compounds are accommodated by this method [118].	Lacks selectivity [119]. Produces lower yields compared to novel methods [120]. Consumes large volumes of water compared to novel methods [121]. Use of water as the solvent risks the development fungal and bacteria, which can degrade bioactive phytoconstituents [112].	Longer duration for the extraction process to allow greater extraction of bioactive phytoconstituents [122]. For delicate plant material such as flowers and leaves, extraction shall be performed at lower than boiling temperatures of the solvent [57]. Covering or uncovering the vessel of extraction, depending on the bioactive phytoconstituents being extracted [122].
**Soxhlet extraction**	Removal of pesticides and organic pollutants from environmental samples [123]. Pharmaceutical industry for bioactive compound extraction. Food industry for flavor and to determine nutritional content [123,124].	The recuring application of the solvent extracts more bioactive phytoconstituents [56]. CO_2_ can be recycled, which alleviates the greenhouse effect. This extraction technique is widely used to extract fragrance-and-aroma oils due to its simplicity. This method requires less solvent than traditional methods. Does not require solvent filtration from the sample following the extraction process [125]. A large quantity of medication may be extracted using a little amount of solvent [108]. Used for extracting heat-stable compounds [58]. High-efficiency, automatic, continuous extraction technology that consumes less time and solvent than maceration or percolation [126]. The Soxhlet extraction method is also a reliable method for the extraction of fat-soluble phytochemicals [92].	Lacks selectivity of heat-sensitive phytoconstituents [125]. Uses hazardous and flammable organic solvents with potential of toxic emissions during extraction [57,127]. Extensive duration of the extraction process [56]. Large volumes of solvent required for optimal extraction [128]. The Soxhlet apparatus is not provided with any kind of agitation or stirrer to facilitate mass transfer [59]. The prolonged extraction process and the high volume of solvents also make this process expensive [129]. Not suitable for samples with high moisture content and roughly ground samples [130,131]. The requirement for ultrapure solvents increases the cost of the procedure [132]. This method is unfavorable for the environment and can cause environmental contamination [133].	Soxhlet rinsing of graphite oxide after Hummers’ oxidation method [134]. Use of a double bypass Soxhlet apparatus to reduce the overall extraction time [135]. High-pressure Soxhlet extraction in which the extractor is placed in a cylindrical stainless-steel autoclave [125]. Automated Soxhlet extraction equipped to perform reflux boiling and Soxhlet extraction followed by extractant recovery [125]. Ultrasound-assisted Soxhlet extraction in an extractor based on Soxhlet’s physiochemical principles [125].
**Accelerated solvent extraction**	This extraction technique is predominantly used in the extraction of fragrance-and-aroma oils [50].	Uses less solvent and requires less time compared to traditional methods [85]. Being a fully automated process, it eliminates possible human errors [136]. Considered an alternative approach to SCFE for the extraction of polar molecules [137].	Susceptible to thermal degradation of heat-sensitive phytoconstituents [60]. Extract impurity due to high pressure applied during extraction [71]. May result in incomplete extractions due to the reduced volume of the solvent during static mode [138]. Costly to run as high pressure is a safety concern [139]. Preparation of extractions for the process is time consuming [140]. Large volumes of solvents are required for rinsing [141]. Sometimes requires the removal of impurities from the extract [141].	Introduction of a fully automated ASE system [87]. Use of non-ionic surfactant solutions Instead of ionic surfactant solutions [142]. Coupling ASE with gas chromatography to extract pyrethroid and organophosphorus residues in herbal plant materials [143].
**Supercritical fluid extraction**	Research and commercial laboratories to produce natural food ingredients, nutraceuticals, and pharmaceuticals and also to remove pesticides from food products [144].	Faster bioactive phytoconstituent transfer rates during extraction compared to traditional methods [77]. SCFE is widely used at the commercial scale. Suitable for extracting thermolabile phytochemicals by keeping the temperature low and the pressure high [145]. CO_2_ is an easily removable solvent from extracts [146]. Near-ambient critical temperature, which reduces a risk of thermal degradation [78]. CO_2_ is non-toxic and a non-flammable solvent [147]. Less polar compounds and small molecules are easily dissolved in super critical CO_2_ [148,149]. Polar compounds and large molecules are easily dissolved with the addition of a co-solvent such as ethanol, methanol, or water [150]. Compounds that are thermally stable as well as high-boiling components can be extracted at low temperatures using the SCFE method [151]. CO_2_ is relatively cheaper when compared to other solvents [152]. The SCF extraction apparatus can be directly connected to gas chromatography for analytical purposes [153]. Reduced extraction duration; suitability for extracting volatile [154]. Reduced solvent use [154]. High pressure in the equipment prevents oxygen entry into the system during extraction, thus preventing oxidation reactions [155]. CO_2_ is considered as environmentally friendly/green solvent [152]. The gas-like diffusivity of supercritical CO_2_ is greater than liquid solvents [156]. Product purity is high, and the decomposition of compounds never happens since a relatively moderate temperature is applied [157,158]. Carbon dioxide can easily be filtered from the extract [158]. The pressure and CO_2_ applied by the system eliminates microorganisms without altering affecting bioactive phytoconstituent composition [147].	Less volatile, polar, and high-molecular-weight compounds are not easily soluble in super critical CO_2_ [148,149]. The need for relatively small sample sizes. Extraction of unwanted compounds occurs [159]. Using an organic modifier (co-solvent) requires an additional purification step in order to remove any remaining solvent [160]. Costly due the high-pressure requirement as it poses a safety concern [161]. Limited ability to solvate highly polar phytoconstituents [162]. Large capital investments are required for the setup of this method’s equipment [161]. The high pressure used in the SCFE operation requires a skilled technician, hence making the extraction costly [161]. Has a limited ability to dissolve fat and water-soluble bioactive phytoconstituents. The equipment is difficult to clean following use.	Addition of polar co-solvents to super critical CO_2_ enhances the polarity and density of the super critical CO_2_ to dissolve polar compounds [150].
**Microwave-assisted extraction**	MAE is used for pharmaceutical and food extractions from plant materials [163].	Shortened extraction time compared to traditional methods [164]. Reduced solvent consumption in relation to traditional methods [164]. In MAE, impurities are removed during extraction. Equipment and setup for MAE costs are relatively modest [165]. Solvent recovery is reasonably high [166]. The use of less polar solvents reduces the risk thermal degradation [167]. Closed extraction consumes less solvent [168]. Closed extraction is best suited for volatile compound extraction [169]. Successful in sample preparation at the laboratory scale [170]. The application of MAE combined with other treatment like ultrasound is recommended in food preservation [171]. Oils extracted with MAE have improved pharmacological properties compared to oils extracted with the Soxhlet extraction method [172]. In comparison to SFE, MAE is a simpler process [165]. In food analysis, MAE can perform condensation and drying using a single piece of equipment [173,174].	High-pressure resistance and air-tightness makes running the extraction costly [175]. Specifically, accommodates solvents that can absorb microwaves [176]. Each cycle only allows for a small quantity of plant material extraction [177]. Volatile solvents are not permitted for use in this method since they degrade the effectiveness of the microwave extraction process [178]. Suitable for relatively smaller molecules [179]. Requires time to cool off to remove the residue after extraction [180]. At industrial-scale utilization of MAE remains very limited [170]. Requires additional clean-up for the removal of the solvent from the sample matrix [180]. Restricted to polar solvent application [181]. MAE efficiency may be very poor when the viscosity of the solvent is high [182]. The microwave extraction of phenol is not efficient as compared to conventional methods [183]. Cannot adequately extract tannins or anthocyanin [184]. MAE has the potential to degrade polyphenols with many hydroxyl-type substituents and heat-sensitive polyphenols such as Anthocyanin [185].	Introduction of modifications such as the pressure and flow of oxygen [85]. Introduction of vacuum microwave-assisted extraction, nitrogen-protected microwave-assisted extraction, ultrasonic microwave-assisted extraction, and dynamic microwave-assisted extraction modifications [186]. Introduction of mechanical or magnetic stirrers [187]. Some articles reported that an enhancement in oil yield under MAE was due to a decrease in viscosity when increasing the temperature [188].
**Sonication/ultrasound-assisted extraction**	This extraction method is used in wastewater purification, oil extraction, and techniques that require cell disruption to obtain intracellular structures from plants, food processing, and diagnostic clinical settings [99,189,190].	Shorter extraction times in comparison to traditional extraction methods [86]. Shorter extraction times and better yields than traditional methods [191]. UAE preserves the composition of bioactive phytoconstituents that degrade at elevated temperatures [192]. Less solvent usage in comparison to traditional extraction methods [86]. The probe system has a shorter extraction time due to lower energy loss to surroundings [97]. Another study reported that UAE is able to reduce the degradation of a thermal-sensitive compound in essential oil [193]. Ultrasonic waves can propagate through any medium [191]. The choice of solvents that can be considered in this type of extraction is wide [94]. Low energy and solvent consumption [194]. Compared to other conventional techniques, the application of the ultrasonic bath in industry is cheaper and quite simpler [189]. In comparison to modern extraction techniques, ultrasonic equipment is less costly and simpler to use [191]. Can extract a wide range of natural compounds using a wide range any solvent [191]. UAE has an extraction efficiency of 85 to 97% depending on the type of sample being extracted at a 20–80 °C temperature range within 10–60 min [191].	Cavitation sometimes degrades plant material, which reduces yield [195]. Lack of uniformity in the process. Direct immersion of the probe into the sample sometimes increases the risk of thermal degradation [98]. Filtration and clean-up step required. The bath system is less efficient due to energy loss to the medium [97]. Furthermore, low reproducibility is also one of the major concerns of the ultrasound bath system [196]. The design data for the application of ultrasound at the industry scale is still limited to date [197]. Due to the action of ultrasound, changes in the extractant medium occur, negatively affecting yield [194,198]. The disadvantages of applying UAE are a decline in power over time and a decline in uniformity in the distribution of ultrasonic energy [191]. An increase in temperature initially improves UAE yield, but as the temperature rises above 45 oC, the yield declines [192].	Combination of sonication with alternative extraction methods—MAE, ASE, and SCFE—to shorten the extraction time [189]. Direct immersion of the probe into the sample causes a rise in temperature in a shorter space of time. Therefore, a jacketed cooling system is required to carry out the extraction process [199]. Using ionic liquids instead of conventional organic solvents assisted by ultrasound extraction improves the process [94].

**Table 2 plants-14-00206-t002:** Widely used South African medicinal plants, their common traditional preparation methods, known active phytoconstituents, and the potential use of post-extraction processing to enhance the activity of phytoconstituents.

Medicinal Plant	Therapeutic Effects	Traditional Preparation Methods	Phytochemical Constituents	Post-Extraction Methods to Increase Efficacy	References
***Adansonia digitata* L.**	Oral infections and dental disorders.	Burning of the plant material to ash	Flavonoids, glycosides, saponins, tannins, phenols, terpenoids, anthraquinones, steroids, reducing sugars, and alkaloids.	No records	[219,220,221,222]
** *Agrimonia eupatoria* ** **Krylov**	Oral infections and dental remedy.	Decoction	Tannins, coumarins, polysaccharides, flavonoids, phenolic acids, and terpenoids.	No records	[223,224,225]
***Aloe ferox* (Mill.)**	Laxative, topical gel/paste, analgesic, antiviral, antiparasitic, antitumor, antimicrobial, antihypertensive, wounds, burns, gastric ulcers, and oedema remedy.	Decoction, pulverized gel extract/paste	Phenolic compounds, flavonoids, alkaloid, tannins, and polysaccharides.	No records	[226,227,228,229]
***Aspalathus linearis* (Burm.f.) Dahlg.**	Vomiting, stomach cramps, immunomodulatory, antiviral, antidiabetic, antihypertensive, antiaging, antieczema, and antimicrobial remedy.	Infusion	Flavonoids, glycosides, phenolic compounds, alkaloids, and polyphenols.	Nanoparticle synthesis	[230,231,232,233,234,235]
***Eucomis autumnalis* (Mill.) Chitt.**	Wound healing, flu, common colds, antihyperglycemic, urinary diseases, stomach aches, fevers, colic, viral and bacterial infections, flatulence, hangovers and syphilis, childbirth, stomach ache, colic, syphilis, fever, urinary diseases, pulmonary ailments, fracture healing, urinary inflammation, oral blisters, and eczema remedy.	Decoctions	Flavanones and terpenoids.	Nanoparticle synthesis	[230,231,236,237,238]
***Harpagophytum procumbens* (DC. ex** **Meisn.)**	Allergies, analgesia, antidiabetic, appetite stimulant, childbirth difficulties, dysmenorrhea, oedema, fever, gastrointestinal disorders, and headache remedy.	Decoction	Glycosides, triterpenes, flavonoids, phenols and flavonoids, phenolic acids, and carbohydrates.	No records	[239,240,241,242,243,244,245]
** *Pelargonium sidoides DC.* **	Treatment of respiratory infections, disorders of the gastrointestinal tract, antibacterial, respiratory tract infections, gastrointestinal disorders, bronchitis, common cold, respiratory ailments, diarrhea, vomiting, and antiviral tract disorders.	Infusion, decoction	Phenolic acid compounds, coumarins, flavonoids, and proanthocyanins.	No records	[246,247,248]
** *Plumbago auriculata* **	Treatment for hyperglycemia, cardiovascular diseases, kidney infections, gastrointestinal disorders, respiratory disorders.	Decoction	Tannins, phenols, alkaloids, saponins, and flavonoids.	Nanoparticle synthesis	[249,250,251]
***Psidium guajava* L.**	Antihyperglycemic and antibacterial effect remedy.	Infusion	Glycosides, polysaccharides, flavonoids, terpenes, tannins, phenols, alkaloids, saponins, and carbohydrates.	Nanoparticle synthesis	[12,252,253,254,255]
***Sclerocarya birrea* (A. Rich.) Hochst.**	Treatment for diarrhea, insect bites, malaria, diarrhea, microbial, plasmodial, hypertensive, diabetic tissue injury.	Decoctions, powdered, infusions	Polyphenols, flavonoids, tannins, steroids, glycosides, flavonoids, alkaloids, and phenols.	Nanoparticle synthesis	[256,257,258,259,260]
***Sutherlandia frutescens* (L.) R.Br.**	Stomach ailments, backache, diabetes, stress, fever, wounds, body rash, bladder, kidney, urinary tract, infection remedy.	Infusion and decoction	Flavonoids, glycosides, phenolic compounds, saponins, and terpenoids.	Nanoparticle synthesis	[230,237,261,262]

## Data Availability

No data was used to support this study.

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
