# Peer review of "Extraction and Processing of Bioactive Phytoconstituents from Widely Used South African Medicinal Plants for the Preparation of Effective Traditional Herbal Medicine Products: A Narrative Review"

_plants, 2025, doi:10.3390/plants14020206_

Round 1

Reviewer 1 Report (Previous Reviewer 3)

Comments and Suggestions for Authors

In my opinion, the revised manuscript is suitable for publication.

Author Response

Reviewer's comment: In my opinion the revised manuscript is suitable for publication

Authors' response: We would like to thank the reviewer for s/her comment

Reviewer 2 Report (Previous Reviewer 2)

Comments and Suggestions for Authors

The manuscript addresses an important topic regarding the extraction and processing of bioactive phytoconstituents from medicinal plants, which holds potential relevance in both traditional medicine and modern therapeutic applications. However, the current version of the manuscript exhibits several critical limitations that must be addressed to enhance its quality and relevance:

  1. Significance and Innovation:

    • While the topic is significant, the manuscript primarily compiles existing knowledge without providing novel insights or innovative contributions. Future revisions should include a deeper critical analysis and novel perspectives, particularly regarding the comparative advantages of traditional versus modern extraction methods.
  2. Methodological Details:

    • The discussion of extraction methods is extensive but lacks a critical comparison of their effectiveness, efficiency, and applicability in industrial versus laboratory contexts. Specific recommendations for different use cases should be included.
    • Address the practical challenges of implementing these methods on a large scale, including sustainability, cost, and reproducibility.
  3. Data and Evidence:

    • The conclusions drawn are not always sufficiently supported by data or references. Incorporating recent and high-quality studies, as well as case studies or examples, would provide stronger evidence for the claims made.
  4. Clarity and Focus:

    • Several sections of the manuscript contain redundant or overly detailed descriptions of extraction techniques that are well-known. A more concise and focused approach would improve the readability and relevance of the text.
  5. Quality of English:

    • The manuscript would benefit from a thorough language review to eliminate redundancies, simplify overly complex sentences, and improve overall readability. Clearer expression of ideas will help communicate the research more effectively.
  6. Practical Implications:

    • The manuscript should discuss how the findings can be applied in real-world contexts, such as industrial production or conservation strategies. This would make the work more impactful and relevant.
Comments on the Quality of English Language

The manuscript is understandable, but the quality of English could be improved to enhance clarity and readability. Redundant phrases and overly complex sentences should be revised for better flow and precision. A professional language editing service is recommended.

Author Response

Comments and Suggestions for Authors

The manuscript addresses an important topic regarding the extraction and processing of bioactive phytoconstituents from medicinal plants, which holds potential relevance in both traditional medicine and modern therapeutic applications. However, the current version of the manuscript exhibits several critical limitations that must be addressed to enhance its quality and relevance:

  1. Significance and Innovation:

    • While the topic is significant, the manuscript primarily compiles existing knowledge without providing novel insights or innovative contributions. Future revisions should include a deeper critical analysis and novel perspectives, particularly regarding the comparative advantages of traditional versus modern extraction methods.
    •  
    • Authors' response: 

      We would like to thank the reviewer for their comments and suggestions to improve the paper. In line with the editor’s comments, we have included a table of most widely used medicinal plants in the preparation of African traditional medicines which we believe strengthens the significance and innovation of the paper.

      Table 1, highlights advantages, disadvantages and suggested inputs to improve each extraction method.
  2. Methodological Details:

    • The discussion of extraction methods is extensive but lacks a critical comparison of their effectiveness, efficiency, and applicability in industrial versus laboratory contexts. Specific recommendations for different use cases should be included.
    • Address the practical challenges of implementing these methods on a large scale, including sustainability, cost, and reproducibility.
    •  
    • Authors' response:
    • We broadly compared the extraction methods in terms of cost, environmental impact, and large-scale feasibility where applicable in Table 1 provided in the paper. Since the review focuses on the pre-extraction processing, extraction and post-extraction processing to enhance the potency and bioavailability of bioactive phytoconstituents from medicinal plants to promote sustainable use of the limited plant resources. In the limitations, we have highlighted the gap between the laboratory and industrial applications of these methods. Comparing these methods in terms of contexts, such as industrial production, clinical medicine, cost, environmental impact, and large-scale feasibility is beyond the scope of this paper.
  3. Data and Evidence:

    • The conclusions drawn are not always sufficiently supported by data or references. Incorporating recent and high-quality studies, as well as case studies or examples, would provide stronger evidence for the claims made.
    •  
    • Authors' response:
    • We have revisited the conclusion to be in line with the stated aim of this review article.
    •  
  4. Clarity and Focus:

    • Several sections of the manuscript contain redundant or overly detailed descriptions of extraction techniques that are well-known. A more concise and focused approach would improve the readability and relevance of the text.
    •  
    • Authors' response:
    • We have taken the reviewers suggestion to seek help from a professional editor to improve the readability of the paper.
  5. Quality of English:

    • The manuscript would benefit from a thorough language review to eliminate redundancies, simplify overly complex sentences, and improve overall readability. Clearer expression of ideas will help communicate the research more effectively.
    •  
    • Authors' response:
    • We have taken the reviewers suggestion to seek help from a professional editor to correct the grammar and spelling used in the paper. We are hopefully that this improved the readability of the article. 
  6. Practical Implications:

    • The manuscript should discuss how the findings can be applied in real-world contexts, such as industrial production or conservation strategies. This would make the work more impactful and relevant.
    •  
    • Authors' response:
    • This study is part of the first authors PhD degree requirements by the institution. Currently we are looking to apply these extraction methods of traditional medicines in a laboratory setting to study their feasible. We can consider the reviewer’s suggestion as a future study.
Comments on the Quality of English Language

The manuscript is understandable, but the quality of English could be improved to enhance clarity and readability. Redundant phrases and overly complex sentences should be revised for better flow and precision. A professional language editing service is recommended.

Authors' response:

We have taken the reviewers suggestion to seek help from a professional editor to correct the grammar and spelling used in the paper. We are hopefully that this improved the readability of the article.

Reviewer 3 Report (Previous Reviewer 1)

Comments and Suggestions for Authors

The authod has revised the MS.

Author Response

Reviewer's comment: The authod has revised the MS

Authors' response: We would like to thank the reviewer for their comment.

Round 2

Reviewer 2 Report (Previous Reviewer 2)

Comments and Suggestions for Authors

The manuscript continues with the deficiencies previously pointed out in the other reviews.

Author Response

Reviewer: The manuscript continues with the deficiencies previously pointed out in the other reviews.

Authors: We would like to thank the reviewer for their comment, on our previous submission we did provide a rebuttal for the comments that the reviewer suggested. We were hoping the reviewer was going to respond to the comments we made regarding why we rebutted some of the comments while some were implemented. 

This manuscript is a resubmission of an earlier submission. The following is a list of the peer review reports and author responses from that submission.

Round 1

Reviewer 1 Report

Comments and Suggestions for Authors

The revised review discusses the importance of traditional herbal medicines, particularly in developing countries, where about 80% of the population relies on them for healthcare. It focuses on the extraction and post-extraction processes to enhance the potency and bioavailability of bioactive phytoconstituents from medicinal plants, promoting sustainable use of these limited resources.

The author has incorporated all the suggestions. Minor adjustments are needed.

Heading 3.2 It seems to be incomplete. Add the rationale of the study.

Heading 4.0

The variability in the composition of phytoconstituents due to seasonal, environmental, and harvesting conditions is another challenge that the passage highlights. However, it does not address how to standardize or mitigate these variations to ensure consistent quality and efficacy of herbal products.

The passage mentions that traditional health practitioners (THPs) rely heavily on wild harvesting, which may face limitations due to urbanization and habitat destruction. While this is noted as a challenge, the text does not propose practical solutions to address the sustainability of wild-harvested resources, which is a critical flaw given the growing global demand for medicinal plants.

Although the review touches on the use of electronic ethnobotanical books and databases, it does not address potential biases or limitations in using these sources. Ethnobotanical knowledge is often context-specific, and broad generalizations could overlook important regional differences in plant usage or preparation techniques.

List the controlling agencies for the herbal drug industry. Provide the web link.

Add references from the current decades. 

Reviewer 2 Report

Comments and Suggestions for Authors

The manuscript has shown improvements in terms of overall clarity and organization, but there are still key areas that require further development to strengthen its contribution to the field. Below are some suggestions to enhance the work:

Deepening the analysis: Although the description of extraction methods has been expanded, the analysis remains superficial. It would be beneficial to provide a more critical and detailed evaluation of the advantages and disadvantages of each method in specific contexts, such as industrial production or clinical medicine. Comparing the methods in terms of cost, environmental impact, and large-scale feasibility would add greater practical value to the manuscript.

Strengthening the conclusions: The conclusions are still limited and do not adequately summarize the key findings. It is important to strengthen this section by highlighting the main points discussed in the text and providing clear recommendations for the herbal medicine industry and future research. Including concrete proposals for improving extraction and processing methods would be very helpful for readers.

Format and coherence: Although the coherence of the text has improved, some formatting errors remain that need to be corrected. For example, the "0" section, which seems to be a generic instruction, should be reviewed and removed to improve the presentation of the manuscript and avoid confusion.

Clarity in writing: Despite improvements in the writing, some sections are still redundant or difficult to follow. We recommend additional editing to improve the flow and make the text more concise and easier to read.

Comments on the Quality of English Language

The manuscript is generally understandable, but there are several areas where clarity could be improved. Some sentences are lengthy and could be simplified for better readability. Additionally, there are minor grammatical issues and redundancies that, if addressed, would improve the overall flow and coherence of the text. A moderate revision of the English language is recommended to make the manuscript more concise and polished.

Reviewer 3 Report

Comments and Suggestions for Authors

The review article “Extraction and processing of bioactive phytoconstituents from medicinal plants for the preparation of effective traditional herbal medicines products: a narrative review”

By Sphamandla Hlatshwayo, Nokukhanya Thembane, Suresh Babu Naidu Krishna, Nceba Gqaleni, and Mlungisi Ngcobo

describes different methods, both classical and recent, to extract phytoconstituents from plants.

The topic is interesting and the manuscript is well structured.

The language must be checked. For example the sentence at rows 67-89, and so on.

All references must be inserted. I noted that the last reference cited in the main text is 190, but in reference section the list arrives to 197!